# MOTION-R1: LATENT-INTENT MOTION GENERATION WITH PHYSICAL CONSISTENCY

## ABSTRACT

Human motion synthesis serves as a foundational component in computer graphics, embodied AI, and robotics. Despite progress has elevated motion quality and physical plausibility, prevailing methods remain constrained by their reliance on explicit and hand-crafted control cues. More importantly, they rarely exhibit the capacity to infer users' implicit intentions—posing a major barrier to human-aligned motion generation. Inspired by DeepSeek-R1's success in eliciting reasoning abilities through rule-based reinforcement learning (RL), we propose **Motion-R1** as the first attempt to explore the R1 paradigm for physically consistent latent-intent motion generation. However, the naïve adoption of Group Relative Policy Optimization (GRPO) to motion synthesis encounters two limitations: (1) *the scarcity of motion-reasoning dataset*, and (2) *a lack of motion reasoning abilities*. Towards these issues, we first construct a newly curated **Motion2Motion** benchmark dataset, comprising text-to-motion dialogues for RL training. Further, our proposed Motion-R1 integrates a JS-divergence constrained policy optimization, achieving improved reasoning capabilities on both motion generation and mathematical computation benchmarks. In addition, we utilize a low-level RL-based optimization strategy to enforce strict adherence to kinematic constraints. Experimental results showcase that Motion-R1 delivers contextually appropriate, lifelike motions and surpasses strong baselines in both accuracy and interpretability. *Code will be released.*

## 1 INTRODUCTION

Human motion generation has garnered significant research attention, with numerous methods proposed to tackle various challenges in this domain Tevet et al. (2022); Zhang et al. (2024a); Dabral et al. (2022); Zhang et al. (2023b); Chen (2024); Andreou et al. (2024); Zhang et al. (2023c); Barquero et al. (2024); Meng et al. (2024). A substantial body of work focuses on synthesizing actions from long textual descriptions, revealing the intrinsic complexity of semantic-to-motion mapping Jing et al. (2023); Jiang et al. (2023); Wang et al. (2024a); Lee et al. (2024); Sun et al. (2024). Nevertheless, existing approaches predominantly address single-turn or isolated commands and lack the capacity to effectively interpret and generate coherent motions from multi-turn or multi-round dialogue inputs. This limitation significantly restricts their applicability in realistic, complex scenarios where contextual continuity and nuanced intent understanding are essential.

Recently, there has been growing interest in bridging the gap between motion generation and its application within physical or simulated environments Cui et al. (2024a); He et al. (2025; 2024); Cheng et al. (2024). However, most Text-to-Motion (T2M) techniques face difficulties in ensuring physical consistency while simultaneously adapting to dynamic environmental constraints and kinematic feasibility, thereby limiting their practicality for deployment beyond controlled simulations.

As illustrated in Fig. 1, prior methods can be broadly divided into two categories: those generating motions without enforcing physical constraints, which often produce visually plausible but physically unrealistic movements; and those incorporating physical constraints but failing to capture the complexity of semantic contexts inherent in multi-turn dialogues. For instance, simple instructions such as "a person is walking around casually" may be reasonably handled by physics-agnostic methods, whereas commands involving detailed postures, gait variations, or context-dependent nuances often lead to motions that are either physically implausible or semantically inconsistent. This dichotomy

exposes a critical research gap, as neither approach sufficiently balances the demands of physical realism and rich contextual comprehension.

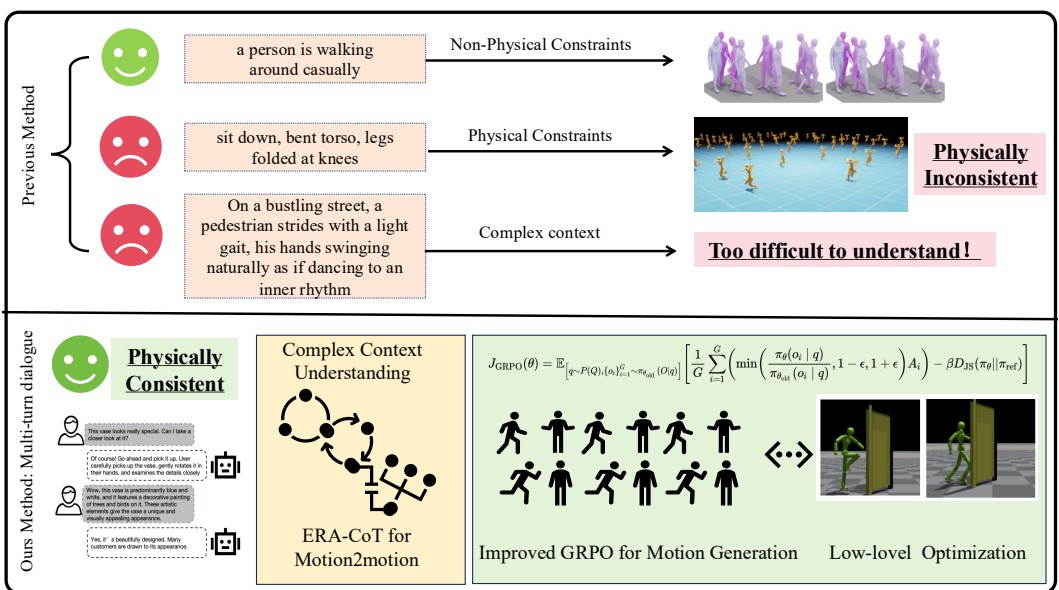

Figure 1: Comparison between prior Text-to-Motion methods and our proposed approach. Existing methods either neglect physical constraints or struggle with understanding complex multi-turn dialogues, resulting in physically inconsistent or semantically inadequate motions. Our approach integrates ERA-CoT for nuanced motion-to-motion reasoning, enhanced GRPO for robust motion policy optimization, and a reinforcement learning-based low-level trajectory refinement to generate physically consistent and semantically coherent motions suitable for deployment in physically constrained simulation environments.

To address these challenges, we propose a novel *text-to-motion policy* generation task, aiming to synthesize motion policies that are both semantically faithful and physically consistent, thereby enabling more realistic application within simulation settings. Our method, **Motion-R1**, contributes in three key aspects: (1) We systematically analyze the effects of semantic ambiguity on motion generation, demonstrating that conventional models often fail to resolve underspecified intentions, resulting in contextually inappropriate motions. (2) We construct a newly curated large-scale **Motion2Motion** (M2M) benchmark dataset consisting of text-to-motion dialogues annotated with latent intent reasoning chains, enabling reinforcement learning-based policy training that integrates a JS-divergence constrained Group Relative Policy Optimization (GRPO) scheme to enhance reasoning and generation capabilities. (3) We design a reinforcement learning-driven low-level optimization framework that explicitly enforces kinematic feasibility and environmental dynamics during motion synthesis within a physically constrained simulation environment, achieving superior performance under such conditions.

## 2 RELATED WORK

### 2.1 HUMAN MOTION SYNTHESIS

How to synthesize realistic human behavior is a long-standing topic. Recent human motion generation research focuses on diffusion models and transformers for diverse, high-quality synthesis. Early methods Barsoum et al. (2018); Kania et al. (2021); Martinez et al. (2017); Petrovich et al. (2022) using GANs Goodfellow et al. (2020)/VAEs Kingma et al. (2013) improve temporal coherence but face mode collapse. Emerging diffusion models and autoregressive models have significantly dominated the field of motion synthesis. The former, *e.g.*, MDM Tevet et al. (2022), MLD Chen et al. (2023), and Tender Wang et al. (2024b) propose conditional diffusion models to learn a powerful probabilistic mapping from texts to motion sequences for controllable text-driven motion generation.

In autoregressive-based methods, Zhang et al. (2023a); Guo et al. (2024); Pinyoanuntapong et al. (2024a;b) embeds human motions into a latent representation space, from which motion sequences are auto-regressively decoded conditioned on textual inputs. More recently, *e.g.*, MotionGPT Jiang et al. (2023); Zhang et al. (2024b), MotionGPT-2 Wang et al. (2024a), M3-GPT Luo et al. (2024), AvatarGPT Zhou et al. (2024), and MotionAgent Wu et al. (2024) have initiated the development of a unified motion-language model aimed at generating plausible human motions alongside along with textual descriptions driven by prompt instructions. However, these methods are typically inferior in physical plausibility and prone to synthesizing motions with artifacts, such as penetration, floating, and sliding. Recent advancements in physics-based methods Peng et al. (2021); Hassan et al. (2023); Xu et al. (2025); Pan et al. (2025); Cui et al. (2024b); Xiao et al. (2023) show promising potential in ensuring physical plausibility through the utilization of physics-aware simulators.

## 2.2 Reward Models for Reasoning

Recent RL-based reasoning frameworks transition from supervised fine-tuning to reward-driven optimization, leveraging auxiliary reward models to evaluate intermediate reasoning quality and enhance generalization Schulman et al. (2017b); Raffel et al. (2020); Christiano et al. (2017). While PPO Schulman et al. (2017a) employs explicit value networks for advantage estimation, value-model-free methods like GRPO Shao et al. (2024) utilize rule-based group-relative mechanisms. Practical software engineering applications demonstrate that continuous rewards via lightweight similarity metrics Liu et al. (2024); Yu et al. (2023); Zhang et al. (2020) outperform binary alternatives by providing granular feedback. Resource-constrained environments reveal trade-offs: small LLMs achieve rapid reasoning gains through RL fine-tuning Team et al. (2024b); Devlin et al. (2019); Hu et al. (2021), albeit with increased computational costs. Emerging paradigms like pairwise preference reward models (PPRM) combine human feedback principles Christiano et al. (2017); Ziegler et al. (2019); Bai et al. (2022); Stiennon et al. (2020) with direct preference optimization techniques Rafailov et al. (2023); Azar et al. (2023); Ethayarajh et al. (2023).

## 2.3 Large Language Models

Fueled by vast datasets and substantial model sizes, Large Language Models (LLMs) represented by GPTs OpenAI (2023b;a), T5 Raffel et al. (2020), PaLM Chowdhery et al. (2023), Gemma Team et al. (2024b; 2025), Qwen Yang et al. (2024a;b), and LLaMA Touvron et al. (2023); Dubey et al. (2024) have recently received extensive attention from researchers for their exceptional abilities showcased in both comprehension and generation task. Earlier models, such as BERT Devlin et al. (2019) and Google T5 Raffel et al. (2020); Chung et al. (2024), were designed for specific tasks like translation or sentiment analysis. The field has since evolved toward general-purpose foundation models, with representative open-sourced LLMs like LLaMA series Touvron et al. (2023); Dubey et al. (2024) and Vicuna family Chiang et al. (2023) have attracted much academic attention. Since these two LLMs are predominantly pre-trained on English corpus, they are limited in multi-language support. Recent years, models like GPT-4 OpenAI (2023a)/ChatGPT OpenAI (2023b), Gemini Team et al. (2024a), Deepseek Liu et al. (2024) benefit from their expansive training datasets (GPT4, for example, about 45 gigabytes) and vast parameter counts, which excel in following natural language instructions and complete real-world tasks. Recent advances in LLM, e.g., DeepSeek-R1 Guo et al. (2025), o3-mini OpenAI (2025), MetaMath Yu et al. (2023) have witnessed a growing emphasis on reasoning capabilities, particularly for complex tasks involving logical deduction, mathematical computation, and multi-step inference. In alignment with this paradigm, Motion-R1 empowers LLMs with strong motion reasoning via reinforcement fine-tuning, supporting both versatile skill learning and physical motion generation.

## 3 Methods

We follow a coherent pipeline that systematically progresses from dataset construction to policy optimization, ensuring each phase builds upon the previous component's capabilities. The framework comprises three synergistic pillars: (1) *Motion2Motion Dataset* construction to capture multi-turn dialog patterns and motion semantics, (2) *Improved GRPO Algorithm* training for enhanced motion description generation, and (3) *Low-Level Kinematic Optimization* to translate textual descriptions into physically plausible motions. This tripartite architecture establishes a closed-loop system

where the dataset informs model training, the optimized model generates motion specifications, and the low-level policy ensures physical realizability in simulation environments. The components' interdependence creates a virtuous cycle-high-quality data enables effective model training, which in turn produces motion descriptions that facilitate physically-consistent policy learning.

## 3.1 MOTION2MOTION DATASET

### 3.1.1 DATASET OVERVIEW

The Motion2Motion Dataset is central to our framework, offering structured conversational data crucial for training motion generation models. It consists of 7,132 annotated human motion samples, as illustrated below. The dataset is designed to support downstream tasks by capturing both explicit action sequences and implicit physical constraints within its annotations.

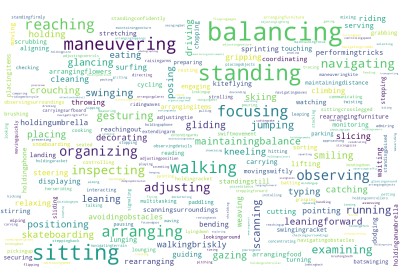 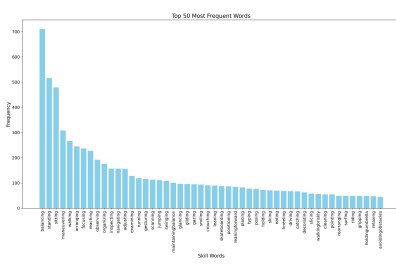

(a) Word Cloud of Generated Skills          (b) Top 50 Frequent Words

Figure 2: Visualizations of Skills and Frequent Words in the Motion2Motion Dataset

To offer a clearer understanding of the skills and frequent words in the Motion2Motion Dataset, we present visualizations in Figure 2. Figure 2a shows a word cloud highlighting the most prominent skills in the dialogues. This visualization provides a snapshot of the key skills frequently discussed. Figure 2b displays the 50 most frequent words in the dataset, offering insights into the common vocabulary used. These visualizations help identify the core concepts and themes, essential for grasping the structure and content of the dialogues.

### 3.1.2 DATASET CONSTRUCTION METHODOLOGY

To ensure both breadth and depth in multi-turn dialogic interactions, we first curated a diverse corpus and used GPT-4 to propose a taxonomic framework that highlights key conversational elements. Domain experts then refined this framework through human-in-the-loop validation, correcting entity–relationship mappings and adding pragmatic nuances that automated methods often miss. Finally, we introduced ERA-CoT (Entity Relationship Analysis with Chain-of-Thought), which decomposes dialogues into explicit and implicit relationships, yielding a corpus that is both ontologically consistent and transferable across domains. The resulting dataset captures surface-level exchanges as well as deeper contextual dependencies, advancing beyond existing resources.

### 3.1.3 ERA-CoT ANNOTATION AND ANALYSIS FRAMEWORK

The ERA-CoT framework is designed to precisely analyze and decompose dialogue structures. It aims to identify hierarchical relationships between entities, enhancing the dataset's ontological consistency and cross-modal transferability for diverse dialogue tasks. ERA-CoT captures both explicit relationships and implicit ones inferred from context. By integrating this framework into dataset construction, we ensured a comprehensive dataset that captures surface interactions while also representing underlying connections and nuances, ultimately creating a more granular and ecologically valid resource for advanced natural language understanding models.

**Entities Extraction:** We identified all relevant entities within the dialogues using the NER (Named Entity Recognition) capabilities of large language models. This step ensured that all significant elements within the dialogues were captured. The entities were then validated using a Self-Consistency (SC) approach, where multiple evaluations were conducted to confirm the accuracy of each entity.

**Explicit Relationship Extraction:** Based on the extracted entities, we identified all directly stated relationships within the dialogues. These relationships were represented as triplets $(e_i, e_j, r)$, where $e_i$ and $e_j$ are entities, and $r$ is the relationship between them. The Self-Consistency method was again employed to ensure the reliability of these relationships.

**Implicit Relationships Inference:** Leveraging the explicit relationships and contextual information, we inferred potential implicit relationships between entities. This step involved generating multiple possible relationships for each pair of entities and scoring them based on their likelihood. The formula for this step can be represented as:

$$R' * i = (e_i, e_j, r_k) \mid i, j \in E \tag{1}$$

where $R' * i$ represents the set of inferred implicit relationships.

**Relationship Discrimination:** To filter out unreliable relationships, we scored each inferred relationship using a scoring agent. Relationships with scores below a predefined threshold $v * th$ were discarded. This step ensured that only the most reliable relationships were retained, enhancing the overall quality of the dataset. The formula for this step is:

$$R_i = (e_i, e_j, r_k) \mid i, j \in E, V(i, j, k) \geq v * th \tag{2}$$

where $V(i, j, k)$ is the confidence score assigned to the relationship $(e_i, e_j, r_k)$.

**Skill Summarization:** Finally, we utilized the refined entities and relationships to summarize the dialogues into coherent and meaningful segments. This step involved integrating the extracted information to generate comprehensive summaries that captured the essence of the dialogues while considering both explicit and implicit relationships.

By following this methodology, we created a high-quality dataset that can effectively support advanced research and applications in dialogue systems and natural language understanding.

## 3.2 ENHANCE GRPO FOR MOTION GENERATION

Building upon the Motion2Motion Dataset's rich annotations, we enhance the GRPO algorithm to specialize in motion description generation. This phase transforms the dataset's structured dialog patterns into a model capable of producing physically-grounded motion specifications. Our improvements specifically address the unique challenges of motion generation, including temporal coherence and kinematic constraint preservation.

### 3.2.1 GRPO ALGORITHM FOR EFFICIENT MODEL TRAINING

The Enhanced GRPO framework capitalizes on the Motion2Motion Dataset's structured entity-relationship annotations through a hierarchical attention mechanism that explicitly models action-semantic interdependencies.

For each input question $q$, the GRPO algorithm samples a group of $G$ outputs $\{o_i\}_{i=1}^{G}$ from the old policy $\pi_{\theta_{\text{old}}}(O|q)$. The optimization objective is then defined as:

$$J_{\text{GRPO}}(\theta) = \mathbb{E}_{[q \sim P(Q), \{o_i\}_{i=1}^{G} \sim \pi_{\theta_{\text{old}}}(O|q)]} \left[ \frac{1}{G} \sum_{i=1}^{G} \left( \min \left( \frac{\pi_\theta(o_i|q)}{\pi_{\theta_{\text{old}}}(o_i|q)}, 1 - \epsilon, 1 + \epsilon \right) A_i \right) \right.$$
$$\left. - \beta D_{\text{JS}}(\pi_\theta \| \pi_{\text{ref}}) \right] \tag{3}$$

Here, $\pi_\theta(o_i|q)$ represents the probability of generating output $o_i$ for question $q$ under the current policy $\pi_\theta$, and $\pi_{\theta_{\text{old}}}(o_i|q)$ is the probability under the old policy. The clipping factor $\epsilon$ controls the range for stable updates, preventing drastic policy changes in regions where the ratio of probabilities could be excessively large. The advantage term $A_i$ is calculated for each output $o_i$, and it quantifies how much better (or worse) an output is compared to the mean of the sampled group. Specifically, the advantage $A_i$ is computed as:

$$A_i = \frac{r_i - \text{mean}(\{r_1, r_2, \ldots, r_G\})}{\text{std}(\{r_1, r_2, \ldots, r_G\})} \tag{4}$$

where $r_i$ is the reward associated with output $o_i$, and the mean and standard deviation are taken over the group of rewards $\{r_1, r_2, \ldots, r_G\}$.

The second term in the objective function involves the Jensen-Shannon (JS) divergence between the current policy $\pi_\theta$ and a reference policy $\pi_{\text{ref}}$:

$$D_{\text{JS}}(\pi_\theta || \pi_{\text{ref}}) = \frac{1}{2} \left[ D_{\text{KL}}(\pi_\theta || m) + D_{\text{KL}}(\pi_{\text{ref}} || m) \right] \tag{5}$$

where $D_{\text{KL}}$ denotes the Kullback-Leibler (KL) divergence, and $m$ represents the midpoint distribution between the two policies $\pi_\theta$ and $\pi_{\text{ref}}$. The JS-divergence term helps ensure that the policy update does not diverge excessively from a reference policy, thus maintaining stability during training. The parameter $\beta$ serves as a hyperparameter that controls the strength of this regularization.

We employ Jensen-Shannon (JS) divergence instead of Kullback-Leibler (KL) divergence for three key advantages. First, JS-divergence's symmetric penalty mechanism (unlike KL's asymmetric approach) enables balanced policy adjustments crucial for structured generation tasks like XML/JSON formatting. Second, its inherent gradient stabilization prevents training instability during early phases, especially when handling irregular outputs. Third, the constrained update dynamics ensure stable convergence while maintaining strict syntactic compliance—vital for high-precision formatting requirements.

In conclusion, our integration of Group-based Reinforcement Policy Optimization (GRPO) with JS-divergence regularization constitutes a theoretically sound and empirically validated approach for large language model fine-tuning. This methodological synergy—simultaneously leveraging batch-level reward signals through group-based optimization while maintaining distributional stability via symmetric divergence measures—addresses both computational efficiency considerations and the structural fidelity requirements inherent in complex generation tasks.

### 3.2.2 REWARD FUNCTION DESIGN

The reward function bridges the dataset's semantic structure with motion generation requirements by: Effective reward shaping plays a pivotal role in reinforcement learning for motion generation, as it directly influences the policy's ability to produce high-fidelity motion sequences. We propose a tripartite reward function comprising three critical dimensions: action precision, skill coherence, and structural compliance. The composite reward function integrates these components as follows:

$$R(\tau) = \underbrace{\alpha R_{\text{action}}}_{\text{Behavioral Fidelity}} + \underbrace{\beta R_{\text{skill}}}_{\text{Contextual Relevance}} + \underbrace{\gamma R_{\text{format}}}_{\text{Syntactic Integrity}} \tag{6}$$

where $\alpha, \beta, \gamma \in \mathbb{R}^+$ denote component weights satisfying $\alpha + \beta + \gamma = 1$.

For each candidate response $r_i \in \mathcal{R}$, we define the action precision reward using a parametric mapping function:

$$R_{\text{action}}(r_i) = \mathcal{S}_{\text{cos}} \left( \Phi_{\text{action}}(r_i), a^\star \right) \tag{7}$$

where $\Phi_{\text{action}} : \mathcal{R} \to \mathbb{R}^d$ is the action embedding operator that maps responses to $d$-dimensional action vectors, $a^\star \in \mathbb{R}^d$ denotes the ground truth action vector, and $\mathcal{S}_{\text{cos}} : \mathbb{R}^d \times \mathbb{R}^d \to [-1, 1]$ represents the cosine similarity metric.

Skill alignment is quantified through semantic embedding comparison:

$$R_{\text{skill}}(r_i) = \frac{1}{|S^\star|} \sum_{s_j \in S^\star} \max_{s_k \in \Phi_{\text{skill}}(r_i)} \mathcal{S}_{\text{BERT}}(s_j, s_k) \tag{8}$$

where $S^\star$ denotes the ground truth skill set, $|S^\star|$ represents the cardinality of the set, $\Phi_{\text{skill}}(\cdot)$ extracts skill embeddings from responses, and $\mathcal{S}_{\text{BERT}}$ computes semantic similarity using pre-trained weights.

Structural compliance reward is enforced via deterministic pattern matching:

$$R_{\text{format}}(r_i) = \frac{1}{2} \cdot \mathbb{I}_{\text{XML-valid}}(r_i) + \frac{1}{2} \cdot \mathcal{S}_{\text{tree}} \left( \Psi(r_i), \Psi^\star \right) \tag{9}$$

where $\Psi(\cdot)$ denotes XML parse tree construction and $\mathcal{S}_{\text{tree}}$ measures normalized tree edit distance in [0,1]. The final response quality score combines component rewards via calibrated aggregation:

$$\mathcal{J}(\theta) = \mathbb{E}_{r_i \sim \pi_\theta} \left[ \sum_{t=1}^{T} \left( \alpha_t R_{\text{action}}^{(t)} + \beta_t R_{\text{skill}}^{(t)} + \gamma_t R_{\text{format}}^{(t)} \right) \right] \tag{10}$$

### 3.3 Low-Level Kinematic and Dynamic Optimization

The final component translates GRPO-generated motion descriptions into executable policies, completing the pipeline from dialog understanding to physical realization. This stage addresses the sim-to-real gap by enforcing dynamic constraints that maintain consistency between textual descriptions and physical plausibility.

This final stage operationalizes the motion plans generated by the GRPO-enhanced model into physically executable policies, creating a closed-loop system that reconciles high-level intent from the dataset with low-level dynamic feasibility. In our approach, we employ a low-level optimization strategy grounded in reinforcement learning to generate motion trajectories that adhere to kinematic constraints and environmental dynamics. This policy is trained to ensure that the generated movements are not only task-compliant but also physically feasible, respecting joint limits and avoiding collisions.

The total reward function $r_t$ at each time step $t$ is composed of two components:

$$r_t = w_G r_G(s_t, a_t, s_{t+1}, g) + w_S r_S(s_t, s_{t+1}) \tag{11}$$

where: $-r_G(s_t, a_t, s_{t+1}, g)$ is the task-specific reward, guiding the agent to achieve the desired goal $g$, $-r_S(s_t, s_{t+1})$ is the style reward, ensuring the motion adheres to the desired movement style, $-w_G$ and $w_S$ are the respective weights for the task and style rewards.

The style reward is derived from an adversarial discriminator $D$, which aims to distinguish between real state transitions observed in expert demonstrations and generated state transitions. The discriminator's objective is to maximize the log-likelihood of correctly classifying real transitions and minimizing the log-likelihood of classifying generated transitions:

$$\mathcal{L}_D = -\mathbb{E}_{(s_t, s_{t+1}) \sim D}[\log D(s_t, s_{t+1})] - \mathbb{E}_{(s_t, s_{t+1}) \sim \pi}[\log(1 - D(s_t, s_{t+1}))] \tag{12}$$

The style reward $r_S(s_t, s_{t+1})$ is then computed as the negative log-probability of the discriminator's output. Formally, it is defined as:

$$r_S(s_t, s_{t+1}) = -\log(1 - D(s_t, s_{t+1})) \tag{13}$$

This reward encourages the policy to generate motions that are indistinguishable from the reference data, thereby capturing the desired style.

The low-level policy is trained using reinforcement learning, where the objective is to maximize the expected cumulative reward over time:

$$J(\pi) = \mathbb{E}_{g \sim p(g)} \mathbb{E}_{\tau \sim p(\tau|\pi, g)} \left[ \sum_{t=0}^{T-1} \gamma^t r_t \right] \tag{14}$$

Here, $\gamma$ is the discount factor, $p(g)$ is the distribution over goals, and $p(\tau|\pi, g)$ is the trajectory distribution under policy $\pi$ for goal $g$. The policy $\pi$ learns to generate motions that not only achieve the task objectives but also exhibit the desired style, facilitating the synthesis of diverse and naturalistic behaviors in physically simulated characters.

## 4 Experiments

We evaluate the proposed fine-tuned Qwen2.5-3B models on action and skill generation tasks, comparing them with both non-fine-tuned variants and strong baselines (Qwen2.5 Yang et al. (2024b), Llama3.2 Dubey et al. (2024)). Model performance is assessed using two divergence-based objectives, Jensen–Shannon (JS) and Kullback–Leibler (KL). Additional experiments on GSM8K (Appendix B) further corroborate the effectiveness of JS divergence.

### 4.1 Action Generation Evaluation

We compare original and fine-tuned models under four metrics: **Semantic Similarity (SS)**, **Keyword Matching Rate (KMR)**, **Information Completeness (IC)**, and **Comprehensive Performance Score (CPS)**.

Table 1 shows that fine-tuning improves all models across metrics. In particular, SS and KMR gains indicate that fine-tuned models generate actions that are semantically closer to the references

while better capturing key entities. Improvements in IC suggest that the generated actions contain more of the necessary details, leading to higher CPS overall. Furthermore, JS divergence consistently outperforms KL, highlighting its advantage in balancing semantic alignment and information coverage.

Table 1: Action generation performance (original vs. fine-tuned).

| Model | SS | KMR | IC | CPS |
|---|---|---|---|---|
| Qwen2.5 3B | 0.1701 | 0.2673 | 0.2911 | 0.1774 |
| Qwen2.5 7B | 0.0330 | 0.1186 | 0.1287 | 0.0616 |
| Llama3.2 3B | 0.1634 | 0.2131 | 0.2309 | 0.1524 |
| Llama3.2 8B | 0.0330 | 0.1186 | 0.1287 | 0.0616 |
| Our (JS) | **0.2178** | **0.3191** | **0.3470** | **0.2176** |
| Our (KL) | 0.2111 | 0.3112 | 0.3388 | 0.2117 |

## 4.2 SKILLS GENERATION EVALUATION

We evaluate skill generation by comparing original and fine-tuned models, using **Jaccard similarity**, **precision**, and **recall** as metrics. Fine-tuning is applied with either KL or JS divergence.

Table 2 summarizes the results. Among the original models, Qwen attains higher Jaccard similarity and precision, while Llama achieves higher recall. Fine-tuning consistently improves performance, with JS divergence yielding the best results across all metrics.

Table 2: Skills generation performance: original vs. fine-tuned (KL and JS).

| Model | Jaccard | Precision | Recall |
|---|---|---|---|
| Qwen2.5 3B | 0.0349 | 0.0564 | 0.0580 |
| Qwen2.5 7B | 0.0199 | 0.0335 | 0.0329 |
| Llama3.2 3B | 0.0579 | 0.0997 | 0.0826 |
| Llama3.2 8B | 0.0199 | 0.0329 | 0.0329 |
| Our (JS) | **0.0616** | **0.0940** | **0.1013** |
| Our (KL) | 0.0531 | 0.0840 | 0.0876 |

Table 3: long text input and skill extraction examples

| Long Text Input | Skill |
|---|---|
| In the suffocating emergency situation, the security personnel quickly assessed the circumstances before taking decisive action. With his body slightly leaning backward, his right leg suddenly exerted force, delivering an impact to the door lock with precisely calculated angle and power. With a crisp cracking sound, the wooden door frame split open, causing the door to swing violently inward, creating a life-saving passage for those trapped inside. This tactical entry technique is known as "forced entry" in special security training and represents a standard procedure for emergency rescue operations in enclosed spaces. | **Kick the Door** |

Here we compare with the previous generation of Anyskill, specifically using the Long Text Input from Table 3 as input, testing both Anyskill and our method. The final results are shown in Figure 3, where we can see that Anyskill cannot understand long text, and therefore cannot perform the knocking action, while our model can effectively understand the knocking action.

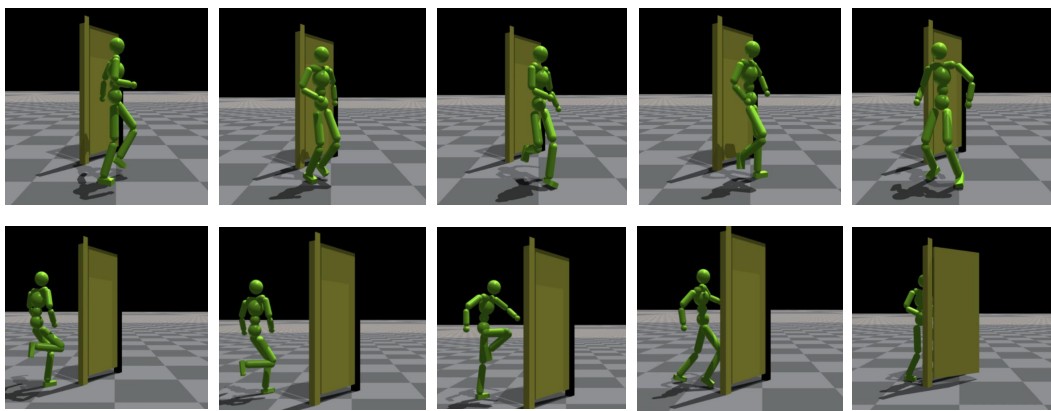

Figure 3: comparison of skill: alternative models (up) vs. our model (low)

### 4.3 ACTION AND SKILL EVALUATION: GPT-4 AS THE JUDGE

In this study, GPT-4 served as an impartial evaluator to assess the rationality and relevance of actions and skills generated by various models, including our proposed model. Evaluations were based on predefined criteria: (1) **Rationality**. Assesses whether actions are contextually appropriate and logically consistent with the given scenario. (2) **Relevance**: Evaluates whether skills align closely with the actions being performed.

Figures 4a and 4b present the comparative evaluation results:

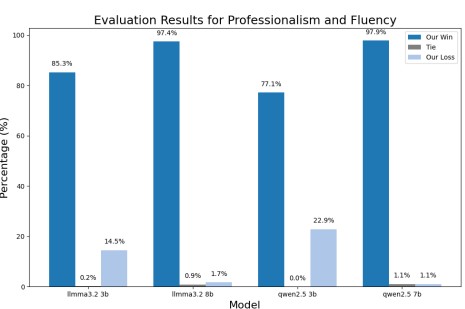

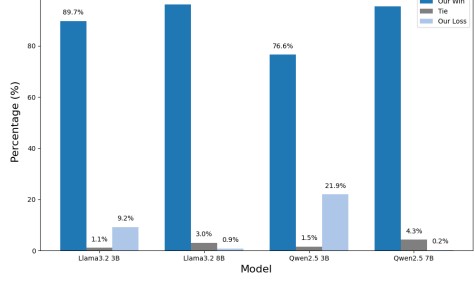

(a) rationality performance comparison

(b) relevance performance comparison

The results demonstrate our model's superior performance in both dimensions. Our model generates more contextually coherent actions that align with common knowledge and situational appropriateness. Additionally, it ensures stronger correlation between listed skills and performed actions, enhancing semantic integrity.

## 5 CONCLUSION

We have presented **Motion-R1**, a novel framework for text-to-motion policy generation that effectively integrates semantic understanding with physical consistency. By utilizing a large-scale **Motion2Motion** dataset with latent intent annotations and employing Generalized Reinforcement Policy Optimization, our method addresses semantic ambiguity in multi-turn dialogues. The reinforcement learning-based low-level trajectory refinement enforces kinematic and environmental constraints within simulated physical settings.

Experimental results show that Motion-R1 surpasses prior approaches in generating motions that are both semantically coherent and physically plausible, advancing the applicability of text-to-motion models in realistic simulations. Future research will focus on real-world deployment and expanding the scope of interaction complexity. Our work provides a foundation for bridging semantic intent and physical feasibility in motion generation.

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

## A    EXAMPLE OF MOTION2MOTION DATASET

```
{
    "action": "a person positioning their surfboard on the sand, scanning
        the waves",
    "skills": [
      "positioning surfboard",
      "scanning waves",
      "standing on sand"
    ],
    "id": "000000417261",
    "conversation": "Before participating in their beach activity, the
        group of people in the image, who have surfboards, should
        consider factors such as the current weather conditions, the surf
         report, and their skill levels. The weather conditions can
        impact the safety and enjoyment of surfing, including factors
        like the temperature, wind direction, and visibility. Checking
        the surf report ensures that the individuals are aware of the
        current and incoming swells and tides, which directly influence
        the quality of the surf. Additionally, the group members should
        assess their skill levels to ensure they can handle the surf
        conditions and avoid putting themselves or others at risk. It's
        also essential to practice proper surf etiquette, such as waiting
         for their turn to catch a wave and respecting others in the
        water. Keeping these factors in mind, the group can maximize
        their enjoyment and safety while participating in the beach
        activity."
}
```

## B    PERFORMANCE ON GSM8K DATASET

Table 4: Quantization Method vs. Divergence Objective on GSM8K

| Configuration | JS Divergence | KL Divergence |
|---|---|---|
| 4-bit Quantized | 0.7263 | 0.7012 |
| 16-bit Full-Precision | 0.8180 | 0.7892 |

