# OpenReview forum: "Motion-R1: Latent-Intent Motion Generation with Physical Consistency"
_ICLR.cc/2026/Conference — Submitted to ICLR 2026_

### Official Review · Reviewer_KPSB · 2025-10-19

**Soundness:** 2
**Presentation:** 2
**Contribution:** 1
**Rating:** 2
**Confidence:** 4

**Summary:**

This paper introduces Motion-R1 as the first exploration of the R1 paradigm for physically consistent, latent-intent motion generation. A straightforward application of GRPO to motion synthesis faces two key challenges: (1) the scarcity of motion reasoning datasets, and (2) insufficient motion reasoning capabilities. This paper curates a new Motion2Motion benchmark comprising text-to-motion dialogues for RL training. Moreover, Motion-R1 employs a JS-divergence–constrained policy optimization, yielding improved reasoning performance on both motion generation and mathematical computation benchmarks. Finally, we incorporate a low-level RL-based optimization strategy to enforce strict kinematic constraints, ensuring physical consistency.

**Strengths:**

1. This paper provides a motion2motion benchmark to do motion reasoning tasks.
2. This paper explore the R1 paradigm for generating human motion with reasoning ability.

**Weaknesses:**

1. The experimental evaluation is insufficient, as it only includes ablations on Qwen2.5 and Llama3.2 without broader model coverage.
2. The paper lacks a direct comparison of physical performance against other motion generation models.
3. Qualitative results are limited, and there is no side-by-side visual or case-based comparison with prior motion generation methods to demonstrate advantages.

**Questions:**

1. Can contemporary LLMs solve this task with strong prompt engineering alone? Please include comparisons against leading models such as GPT-4 and Gemini 2.5.
2. What is the source of the human motion data in the Motion2Motion dataset? Is it derived from AMASS or another dataset? Please clarify it.
3. Does the method support longer-duration human motion generation? How long can the model generate?

---

### Official Review · Reviewer_XmcE · 2025-10-21

**Soundness:** 1
**Presentation:** 1
**Contribution:** 1
**Rating:** 2
**Confidence:** 4

**Summary:**

The paper proposes **Motion-R1**, a text-to-motion framework that: (i) introduces a new “Motion2Motion” (M2M) dataset of **7,132** multi-turn motion dialogues with “latent-intent” annotations; (ii) fine-tunes a small LLM via **GRPO** with a **JS-divergence** regularizer to improve “reasoning” for motion description generation; and (iii) adds a **low-level RL** stage to enforce kinematic/physical consistency. Experiments report improvements on bespoke text metrics (semantic similarity, keyword match, info completeness) and a “GPT-4 as judge” assessment, plus a small GSM8K ablation for JS vs. KL. The work repeatedly asserts physically consistent, semantically aligned motion generation suitable for simulation.

**Strengths:**

* **Problem framing:** Multi-turn “latent intent” is a timely angle for motion generation.
* **Clear optimization objective:** GRPO + a JS penalty is sensible for stabilizing updates against a reference policy.
* **Attempted phys-layer:** A low-level RL controller with adversarial style reward is a reasonable blueprint (akin to AMP-style priors).

**Weaknesses:**

1. **Core claim not demonstrated.**
   The paper claims “physically consistent” generation, yet no kinematic/physics metrics, simulators, or contact/penetration analyses are reported. The “low-level kinematic optimization” (and there is even a typo in Figure 1) is described, but there’s no empirical section showing it running, no environments, no skeleton, and no penetration evaluations—only text-side metrics and a GPT-4 judgment. This is a mismatch between claims (Intro/Conclusion) and evidence (Experiments).

2. **Evaluation is largely text generation, not motion generation.**
   Reported metrics are SS/KMR/IC/CPS and “GPT-4 as the judge.” There are no standard T2M metrics (e.g., R-Precision, FID, MMDist, Diversity), no comparisons to strong T2M baselines on public datasets, and no user or expert studies on motion realism/physicality.

3. **Suspicious/duplicated baseline numbers.**
   In Table 1, Qwen2.5-7B and Llama3.2-8B have **identical** scores across all four metrics. That coincidence across two different model families and sizes is highly implausible and suggests **copy-paste or placeholder values** rather than measured results. Also, intutively speaking, model with larger parameter size usually performs better in such tasks but an opposite pattern is observed in the results provided. The authors did not explain these phenomenon explicitly.

4. **Dataset construction lacks crucial detail and evidence of quality.**
   M2M is said to be curated with GPT-4, NER, self-consistency, and an ERA-CoT pipeline; yet the paper shows **only one** JSON sample (“surfboard on sand”) without actual “latent-intent reasoning chains,” no annotation protocol, no inter-annotator agreement, no quality control, and no release details. As presented, it’s hard to judge dataset validity or contamination.

5. **Reward design refers to a “ground-truth action vector” and skill set but the dataset provides text.**
   The reward defines cosine similarity to a ground-truth action vector ($a^\star$) and set-wise skill similarity, yet the dataset shows *text* fields (“action”, “skills”). The paper doesn’t specify how ($a^\star$) is constructed, nor how action embeddings are calibrated or validated against motion. There is a gap between text rewards to actual motion quality.

6. **Misaligned comparison.**
   The “Anyskill” comparison hinges on a long text paragraph and claims Anyskill “cannot understand long text” and therefore “cannot perform the knocking action,” while their model can. While Table 3 and Figure 3 demostrates their model is able to understand the “kick the door” skill but paragraph conflates “kick the door” with “knocking”.

7. **Missing experimental specifics.**
   The missing experimental details: data splits, prompt templates, and GRPO hyper-parameters (G, $\beta$, $\epsilon$) raises the concern for reproducibility.

**Questions:**

1. In Table 1, two different baselines appear to have identical scores across all metrics. Please confirm whether this is a typesetting error or an evaluation duplication.

2. The reward references a “ground-truth action vector $a^\star$ and “skill similarity.” How exactly is $a^\star$ obtained from text? What embedding model or mapping is used, and how is it calibrated to motion features?

---

### Official Review · Reviewer_wx6E · 2025-10-31

**Soundness:** 2
**Presentation:** 2
**Contribution:** 2
**Rating:** 2
**Confidence:** 3

**Summary:**

This paper introduces Motion-R1, a framework for text-to-motion policy generation that jointly addresses semantic intent extraction from multi-turn user dialogues and physical consistency in motion synthesis. Motion-R1 leverages a new Motion2Motion dataset with richly annotated text-to-motion dialogues, integrates a group relative policy optimization (GRPO) algorithm regularized by Jensen-Shannon divergence, and enforces low-level kinematic feasibility via reinforcement learning-based trajectory optimization.

**Strengths:**

Clear motivation to handle latent intent and multi-turn dialogue in text-to-motion, a practically relevant gap.

Introducing JS-regularized GRPO for structured outputs is reasonable and well-motivated.

The ERA-CoT annotation pipeline for multi-entity/relationship reasoning is interesting and potentially useful.

**Weaknesses:**

- Writing clarity and method specificity

It is unclear what exactly the fine-tuned Qwen outputs as the “target” for motion control. Despite using Rformat (XML-validity and tree similarity), the paper does not provide a concrete schema, examples, or field definitions for the structured output, making it hard to reconstruct a usable high-level specification.
The interface from text to the physical policy is under-specified. The parsing/mapping from structured text to low-level targets g (e.g., gait parameters, keyframes, contact schedules, end-effector trajectories) and the corresponding constraints are not clearly described.
Section 3.3 largely restates standard AMP/adversarial imitation details without presenting method-specific innovations that tightly couple the proposed high-level structured outputs to the low-level controller.

- Missing a clear pipeline figure

There is no pipeline diagram showing the full flow (multi-turn dialogue → structured output → parser/mapping → low-level RL control → physics execution). The absence of a schematic significantly reduces readability and reproducibility.

- Lack of videos and quantitative motion results

As a motion paper, the absence of any video demonstrations or quantitative kinematic/physical metrics is close to unacceptable. Claims of physical consistency and deployability in constrained environments require visual and numerical evidence.

- Limited baselines and evaluation scope

Comparisons are restricted to base LLMs (Qwen/Llama, original vs. fine-tuned) and JS vs. KL ablations. Metrics are text-level (SS/KMR/IC/CPS, Jaccard/Precision/Recall, GPT-4 judging).
There are no quantitative comparisons against standard text-to-motion baselines (e.g., MDM, MLD, TEMOS, T2M-GPT, MotionGPT/MotionGPT-2, Tender, AvatarGPT) or physics-based T2M methods (PADL/SuperPADL/ClosD/PhysDiff/MoConVQ).
The paper provides only a conceptual figure (Fig. 1) contrasting “non-physical” vs. “physical but semantically weak” approaches. It does not report results on shared benchmarks with standard metrics such as FID/DTW, Foot Sliding Rate, Contact F1, MPJPE/MPJVE, ACC/Jerk, penetration rate, fall rate, energy, or task success.

**Questions:**

What is the exact structured schema (JSON/XML) produced by Qwen? Please provide a complete template, field explanations, and an example mapping to low-level targets (phases/contacts/keyframes/heading/gait).

Which kinematic and physical constraints are explicitly encoded at the low level? Can you provide ablations showing the impact of no-slip/contact consistency/keyframe tracking on final motion quality?

Can you provide a sufficient number of video demos and a project page?

---

### Official Review · Reviewer_nF59 · 2025-10-31

**Soundness:** 2
**Presentation:** 1
**Contribution:** 2
**Rating:** 2
**Confidence:** 5

**Summary:**

This paper introduces **Motion-R1**, a framework designed to generate physically consistent human motion from text, specifically claiming to handle multi-turn dialogues. The authors argue that existing methods struggle to understand implicit intent and lack physical realism. To address this, the paper's main contributions are: 1) The construction of a new text-to-motion dialogue dataset, Motion2Motion (M2M); 2) An improved reinforcement learning-based fine-tuning method, GRPO-JS, for reasoning about and generating textual descriptions of actions and skills from dialogue; and 3) A low-level RL optimization strategy to execute the motion in a simulator, ensuring it adheres to kinematic constraints.

**Strengths:**

- **Valuable Task Formulation**: The paper attempts to tackle an important and challenging problem. Translating high-level, potentially ambiguous textual intent (especially from a dialogue context) into a concrete, physically plausible low-level motion policy is a key step toward more natural human-computer interaction and embodied AI.

- **Framework Design**: The two-stage decomposition of this complex task—a high-level policy (GRPO-JS) for semantic understanding and text generation, and a low-level policy for physical execution—is a logically sound design.

- **New Dataset**: The paper contributes a new dataset (M2M) focused on extracting motion intent from dialogue. It could be valuable to the community.

**Weaknesses:**

- **Unfair Comparison**: The paper's experiments fail to prove the superiority of the proposed GRPO-JS. The quantitative evaluations compare a model fine-tuned on M2M (Ours) against base models (Qwen, Llama) that were not fine-tuned. A fine-tuned model will inherently perform better, making this conclusion trivial. The authors need to compare the performance of GRPO-JS against other fine-tuning methods, such as PPO, on the same dataset.

- **Insufficient Experiments**: As a motion generation model, the paper fails to experiment on standard motion generation benchmarks (e.g., HumanML3D). It also does not use standard metrics like FID or R-precision. Only one qualitative case is provided (Fig 3), which is insufficient to prove that Motion-R1 can generate high-quality human motion.

- **Unclear Presentation**: The paper claims to solve the "multi-turn dialogue" problem, but the dataset example provided (Appendix A) shows a single text description. Furthermore, the paper does not explain the connection between GRPO and RL, and the details of how GRPO-generated motion descriptions are translated into executable policies are completely omitted.

**Questions:**

- What was the rationale for comparing GRPO-JS against non-finetuned base models instead of other finetuning methods (like PPO or SFT) on the M2M dataset, which would be a more direct test of the method's superiority?

- Given the paper's focus on motion generation, why were standard benchmarks (e.g., HumanML3D) and metrics (e.g., FID, R-precision) omitted in favor of text-based evaluations?

- Can the authors clarify what defines a "multi-turn dialogue" in the M2M dataset? The example in Appendix A seems to be a single paragraph.

- Could you please detail the mechanism for translating the textual descriptions from GRPO-JS (Sec 3.2) into the goal vector g required by the low-level policy (Sec 3.3)? This critical link appears to be missing.

---

### Meta-Review · Area_Chair_WYrH · 2026-01-07

**Summary:**

The reviewers consistently questioned whether the paper demonstrated its main claims, noting that the evaluation relies mostly on text metrics rather than motion quality, lacks comparisons to standard benchmarks and baselines and omits critical details on data, methodology and reproducibility. The authors did not respond in the discussion phase, so all concerns remain unresolved. I recommend rejection.

**Reviewer Concerns:**

None of the concerns raised in the reviews were resolved, as the authors did not engage in the discussion or provide additional clarification. Without rebuttal or follow-up evidence, all reviewers’ criticisms stand unchanged.

**Reviewer Scores:**

All scores likely remain unchanged, resulting in consistent rejection.

---

### Decision · Program_Chairs · 2026-01-26

Reject